# Comparative Study of Crystallization, Mechanical Properties, and In Vitro Cytotoxicity of Nanocomposites at Low Filler Loadings of Hydroxyapatite for Bone-Tissue Engineering Based on Poly(l-lactic acid)/Cyclo Olefin Copolymer

**DOI:** 10.3390/polym13223865

**Published:** 2021-11-09

**Authors:** Farzana Nazir, Mudassir Iqbal

**Affiliations:** Department of Chemistry, School of Natural Sciences, National University of Science and Technology (NUST), Islamabad 44000, Pakistan; farzana.nazir@sns.nust.edu.pk

**Keywords:** poly(l-lactic acid), cyclo olefin copolymer, hydroxyapatite, cytotoxicity, mechanical properties, bone-tissue engineering

## Abstract

A poly(l-lactic acid)/nanohydroxyapatite (PLLA/nHA) scaffold works as a bioactive, osteoconductive scaffold for bone-tissue engineering, but its low degradation rate limits embedded HA in PLLA to efficiently interact with body fluids. In this work, nano-hydroxyapatite (nHA) was added in lower filler loadings (1, 5, 10, and 20 wt%) in a poly(l-lactic acid)/cyclo olefin copolymer10 wt% (PLLA/COC10) blend to obtain novel poly(l-lactic acid)/cyclo olefin copolymer/nanohydroxyapatite (PLLA/COC10-nHA) scaffolds for bone-tissue regeneration and repair. Furthermore, the structure-activity relationship of PLLA/COC10-nHA (ternary system) nanocomposites in comparison with PLLA/nHA (binary system) nanocomposites was systematically studied. Nanocomposites were evaluated for structural (morphology, crystallization), thermomechanical properties, antibacterial potential, and cytocompatibility for bone-tissue engineering applications. Scanning electron microscope images revealed that PLLA/COC10-nHA had uniform morphology and dispersion of nanoparticles up to 10% of HA, and the overall nHA dispersion in matrix was better in PLLA/COC10-nHA as compared to PLLA/nHA. Fourier transformation infrared spectroscopy (FTIR), powder X-ray diffraction (XRD), and differential scanning calorimetry (DSC) studies confirmed miscibility and transformation of the α-crystal form of PLLA to the ά-crystal form by the addition of nHA in all nanocomposites. The degree of crystallinity (%) in the case of PLLA/COC10-nHA 10 wt% was 114% higher than pure PLLA/COC10 and 128% higher than pristine PLLA, indicating COC and nHA are acting as nucleating agents in the PLLA/COC10-nHA nanocomposites, causing an increase in the degree of crystallinity (%). Moreover, PLLA/COC10-nHA exhibited 140 to 240% (1–20 wt% HA) enhanced mechanical properties in terms of ductility as compared to PLLA/nHA. Antibacterial activity results showed that 10 wt% HA in PLLA/COC10-nHA showed substantial activity against *P. aeruginosa*, *S. aureus*, and *L. monocytogenes*. In vitro cytocompatibility of PLLA/COC10 and PLLA nanocomposites with nHA osteoprogenitor cells (MC3T3-E1) and bone mesenchymal stem cells (BMSC) was evaluated. Both cell lines showed two- to three-fold enhancement in cell viability and 10- to 30-fold in proliferation upon culture on PLLA/COC10-nHA as compared to PLLA/nHA composites. It was observed that the ternary system PLLA/COC10-nHA had good dispersion and interfacial interaction resulting in improved thermomechanical and enhanced osteoconductive properties as compared to PLLA/nHA.

## 1. Introduction

Bone injury is considered one of the major health concerns worldwide and researchers in various fields are working on bone tissue for its functional recovery and regeneration [1]. Naturally, bone tissue possesses the capacity to regenerate damaged bones, such as a crack, and a certain type of fractures. However, bone defects greater than 2 cm cannot be healed without proper surgical treatment [2,3]. Previously, to fix the bones, biologically inert metallic devices, such as anchors and screws, stitches, or bone autografts, allografts have been used for the regeneration of bone tissues [4,5,6]. The problem associated with metal bone fixation is that it requires repeated surgeries; the use of bone allografts might transfer diseases from the donor to acceptor; and the use of autografts can face donor site morbidity and infection during the healing process. Moreover, bone grafts could create severe risks in aging individuals [7]. The advancement in scientific research in bone-tissue engineering has paved the way to replace conventional treatment methods by noninvasive and artificial functional biomaterials. Therefore, a plethora of biomaterial has been introduced as an alternative solution for bone injury recoveries. Biomaterial whose optimized mechanical properties match with that of natural bone can reduce stress and strain imbalance and induce the reorganization of functional tissues, thereby facilitating the bone regeneration process at the injured site [8]. Several biodegradable polymers, such as collagen, cellulose, [9], polyvinyl alcohol (PVA), poly(l-lactic) acid (PLLA), poly caprolactam (PCL), and poly(glycol-c-lactic acid) (PLGA), have been utilized for biomedical and tissue engineering applications [10,11,12].

Bone at the nano level comprises collagen (a polymer) and hydroxyapatite (calcium phosphate). Among all biopolymers, PLLA has emerged as the most promising Food and Drug Administration (FDA) approved polyester for bone-tissue engineering applications [13]. PLLA has unique properties, such as biocompatibility, biosorption, and biodegradability. Moreover, the mechanical stiffness of PLLA can be tuned according to desired applications. However, PLLA has low bioactivity and a lack of osteoconductivity, which limits its usage in bone-engineering applications. One method to improve the osteoconductivity of PLLA is the incorporation of the bone mineral known as hydroxyapatite (HA) as a filler in the polymer matrix. HA is a biocompatible material found in the morphology of bones and has suitable osteoconductive properties for promoting bone regeneration. In vivo trials of nanohydroxyapatite (nHA) for bone-tissue engineering have revealed its ideal biomaterial properties and excellent bone-bonding ability [14]. Moreover, nHA has intrinsic hydrophilic properties essential for protein absorption during bone regeneration, which inhibits the apoptotic flexibility of cells, enhances pre-osteoblasts adhesion, proliferation, and osteogenic differentiation. However, pristine nHA has certain drawbacks to be used as a bone scaffold. Pristine nHA shows agglomeration, which could migrate from its site of introduction, and it is difficult to shape powdered nHA into a form to use as a scaffold. Therefore, the addition of nHA to biodegradable polymers might be a promising approach for the development of bone-tissue engineering scaffolds. nHA, an inorganic phase, when dispersed in a polymer matrix (organic phase), results in reinforcing the polymer matrix, increasing the polymer’s bioactivity and osteoconductivity [15]. Recently, Chen et al. prepared collagen/HA scaffolds to assess biological response and found that these polymeric composites exhibited promising cell recruitment and cell viability and encouraged osteoblast differentiation in vitro and in vivo [16]. Qurat Ul ain et al. synthesized cyclo olefin copolymer/nanohydroxyapatite (COC/nHA) blends to examine cytocompatibility of pre-osteoblasts and found that the cell proliferation rate had been substantial enough to be used as bone-tissue engineering scaffold [17]. Akindoyo et al. prepared poly(l-lactic acid)/nanohydroxyapatite (PLLA/nHA) nanocomposites to study the cell attachment and viability of osteoblasts on the PLLA/HA nanocomposite [18]. Intriguingly, it was found that in PLLA/nHA nanocomposites, nHA aids to neutralize the acidic environment created by lactic acid during degradation of PLLA, which made PLLA/HA composites better for bone regeneration as compared to PLLA alone for the bone scaffold. Furthermore, MgO, ZnO, and growth factors have been incorporated in PLLA/HA to enhance the viability, proliferation, and migration of bone cells [19,20,21].

Although PLLA/HA nanocomposites showed good biocompatibility for bone-tissue engineering applications, it has demerits, such as the lack of tunable mechanical properties, brittleness of PLLA, and poor bonding between nHA and PLLA. Therefore, a ternary system has been introduced to improve the mechanical properties of PLLA/HA composites. For instance, Eftekhari et al. and Cecan et al. designed a ternary scaffold based on PLLA/cellulose/HA nanocomposites for bone-tissue engineering [22,23]. Eftekhari et al. found that crystallinity of nanocomposites increased from 50 to 80% and observed significant enhancement of young modulus from 6.6 MPa to 38 MPa. In another report, de Siqueira et al. introduced PCL as a third phase in PLLA/HA to improve the mechanical properties and behavior of osteoblasts [24]. However, to prepare triphase nanocomposites, sodium dodecyl sulfate (SDS) is used to achieve uniform dispersion of a hydrophobic third polymer in PLLA/HA, but SDS was shown to be toxic to cell viability [25]. Thus, there is always a need for the development of a scaffold with improved tissue-engineering applications, as well as better mechanical properties, without the use of SDS.

To achieve the aforementioned objective, we strategically introduced cyclic olefin copolymer (COC) to PLLA, prepared PLLA/COC blends to improve the mechanical properties of PLLA and checked its compatibility for bone-tissue engineering. COC, an amorphous polymer, provides good mechanical properties [26], antibacterial and antifungal properties, as well as excellent bone-tissue engineering properties [27,28]. So, the introduction of COC in PLLA at 10 wt% resulted in a PLLA/COC10 blend. PLLA/COC10 showed excellent mechanical properties, displayed high crystallinity (as compared to PLLA), and proved potential biomaterial for bone-tissue engineering [11]. In the present study, we prepared nHA intending to introduce nHA at different filler loadings in the PLLA/COC10 blend. A ternary system was expected to be a better substitute for bone-tissue engineering applications as compared to a PLLA/nHA binary system. We evaluated the structure and properties relationship (morphology, crystallinity, and mechanical properties) in the nanocomposites. We compared the structural and thermomechanical properties of PLLA/COC10-nHA and PLLA/nHA and concluded that PLLA/COC10-nHA had better structural and thermomechanical properties Moreover, swelling, degradation, and in vitro studies of the PLLA/COC10-nHA nanocomposites with two different cell lines MC3T3E1 and BMSC were evaluated to see the potential of the scaffolds for bone-tissue engineering applications/bioimplants. PLLA/COC10-nHA showed better cell viability, as well as proliferation, as compared to PLLA/nHA.

## 2. Materials and Methods

### 2.1. Materials

PLLA opaque pallets having an average molecular mass of 100,000 g/mol suitable for bone-tissue engineering were purchased from Polysciences Inc., Philadelphia, PA, USA. COC TOPAS^®^ 8007 amorphous polymer was obtained from Ticona, Sulzbach, Germany. Calcium nitrate tetrahydrate (Ca(NO_3_)_2_·4H_2_O), reagent grade, molecular weight 236.15 g/mol assay (complexometric assay 99–103%), di-ammonium hydrogen phosphate ((NH_4_)_2_HPO_4_), reagent grade, molecular weight 132.06 g/mol (98%), and ammonia solution 35% NH_3_ sol, certified AR for analysis, d = 0.88, Fisher Chemicals, Hampton, NH, USA were acquired and used as such. Dulbecco’s Modified Eagle’s Medium (DMEM), Trypsin-EDTA (0.05%), phenol red, LIVE/DEAD^®^ Viability/Cytotoxicity Kit (mammalian cells) and fetal bovine serum (FBS) were procured from (Gibco) Thermo Fisher Scientific Inc, Waltham, MA, USA. Chloroform, research grade (99% pure), penicillin-streptomycin (10,000 U/mL), Triton X-100, rhodamine phalloidin (Alexa Fluor 594-labeled, Invitrogen™, Waltham, MA, USA), 4% paraformaldehyde (PFA), bovine serum albumin (BSA), 4′,6-diamidino-2-phenylindole (DAPI), and phosphate buffered saline (PBS) were procured from Sigma-Aldrich, Burlington, MA, USA. Two frozen cell lines: frozen MC3T3-E1 (pre-osteoblast cell lines) and BMSC (bone marrow-derived mesenchymal stem cells), were obtained by the courtesy of ATCC, Manassas, VA, USA.

### 2.2. Synthesis of the Hydroxyapatite Nanoparticles

Hydroxyapatite nanoparticles (nHA) synthesis was carried out by combining calcium (calcium nitrate tetrahydrate (Ca(NO_3_)_2_·4H_2_O) and phosphorus (diammonium hydrogen phosphate (NH_4_)_2_HPO_4_)) sources. Reaction mixture was maintained at pH 12 using NH_3_ sol [16]. Briefly, aqueous solutions of Ca(NO_3_)_2_·4H_2_O (dissolving 32.8125 g in 200 mL deionized water at 80 °C) and (NH_4_)_2_HPO_4_) (9.5 g in 137.5 mL in deionized water at 80 °C) were prepared on a magnetic hot plate. Two solutions were mixed by drop-wise addition of Ca(NO_3_)_2_·4H_2_O solution into a highly basic (NH_4_)_2_HPO_4_) solution for 2 h at 80 °C, followed by 24 h of aging. After aging, nHA precipitates were filtered and washed with deionized water. Residue nHA was vacuum oven-dried at 80 °C for 24 h. A muffle furnace was used for calcination at 500 °C for 2 h. The chemical reaction for the synthesis of nHA is described in Equation (1):Ca(NO_3_)_2_·4H_2_O + 6(NH_4_)_2_HPO_4_ + 8NH_4_OH Ca(PO_4_)_6_(OH)_2_ + 20NH_4_NO_3_ + 46H_2_O(1)

### 2.3. Fabrication of Nanocomposites PLLA/COC10-nHA

Neat PLLA and COC were dried in a vacuum oven for 5 h to remove any moisture content. PLLA (0.90 g) in 20 mL of chloroform and COC (0.10 g) in 10 mL of chloroform were dissolved separately to get clear solutions. Two dissolved polymers were combined to obtain the blend. Meanwhile, nHA in different weight percentages (1 wt%, 5 wt%, 10 wt%, and 20 wt%) in 10 mL of chloroform was dispersed for 2 h. Dispersed nHA and the polymer solution were mixed according to the Appendix A, and this mixture was probe sonicated for half an hour for homogenous dispersion of the nHA in the PLLA/COC10. This hybrid nanocomposite solution was stirred for 24 h, and the nanocomposite was air-dried at room temperature for 24 h in glass petri dishes. Air-dried nanocomposite 2D Scaffolds or films were dried in a vacuum oven for 5 h at 300 mbar. These 2D Scaffolds or films were cut into desired shapes for structural, thermomechanical, and cell-culture analysis. Average film thickness varied between 100 mm and 125 mm. The compositions of the solutions prepared are given in Appendix A and fabrication is shown in Appendix A.

### 2.4. Fabrication of Nanocomposites PLLA/nHA

A solvent-casting method was used for the fabrication of the PLLA/nHA composite material. nHA in different weight percentages (1 wt%, 5 wt%, 10 wt%, and 20 wt%) was probe sonicated in a 10 mL solution in chloroform. Neat PLLA in different weight percentages (99 wt%, 95 wt%, 90 wt%, and 80 wt%) was dissolved in 10 mL solution in chloroform. Two solutions were mixed according to Appendix A, then stirred for 2 h. The resulting nanocomposite was poured into a clean petri dish. After 24 h of air drying of the nanocomposite solution in a glass petri dish, films were obtained. These nanocomposite films were vacuum oven-dried for 5 h at 300 mbar. These films were cut into desired shapes for structural, thermomechanical, and cell culture analysis. Average film thickness varied between 100 mm and 120 mm.

### 2.5. Characterization Techniques

Morphological analysis of the nanocomposites was captured by using JSM-6490LV-JEOL, JEOL USA, Inc., Peabody, MA, USA. Nanoparticle morphology and size were observed by placing nHA powder on carbon tape, then directly gold sputtering the powdered nanoparticles. Nanoparticles were also suspended in deionized water, then after water evaporation, sample was gold sputter-coated for SEM analysis. Cryofractured morphology of polymer, blend, and nanocomposites films was obtained by dipping in liquid nitrogen and fracturing the films to obtain fresh cryofractured surfaces. Films were fixed on an aluminum stub, then gold sputtered before performing SEM analysis. Various magnifications below 10,000× due to the polymer cracking at higher magnifications were used to obtain the morphology of nanocomposites.

WAXD patterns for nanoparticles were carried out by Siemens D5005 X-ray Diffractometer (Munich, Germany), equipped with Cu/Kα (1.54060 nm) radiation source. 2*θ* values were varied from 5°–70° at 25 °C. For hydroxyapatite, average crystallite size was calculated by using Scherer’s formula shown below considering Bragg peaks (211) and (310).
$$D=\frac{k\lambda }{\beta \cos\left(\theta \right)}$$
where *D* represents crystal size, *k* = shape factor (0.008), *λ* = incident wavelength, *β* = broadening of the peak at half of the maximum peak, and *θ* is the diffraction angle.

WAXD patterns of PLLA/COC/nHA and PLLA/nHA were measured at a rate of 2°/min between 2*θ* range 5°–70°. During all experiments, voltage was 20 kV and current was at 5 mA.

FTIR spectra of nanocomposites were collected using a Bruker platinum ATR model Alpha spectrophotometer, Mannheim, Germany over an observation range of 400–4000 cm^−1^.

The thermal behavior of nanocomposites was carried out on a DSC 6000 from Perkin Elmer, Foster city, CA, USA. by adopting a melt crystallization route. Nitrogen gas was purged at 20 mL/min in an Intracooler as a refrigerating system. Zinc and indium standards were used to calibrate the instrument before carrying out any experiments. A sample weighing > 6 mg was taken in stainless steel pans to avoid thermal lag. To remove thermal history, nanocomposites were subjected to heat at the rate of 10 °C per min from −40 °C to 250 °C and held at 250 °C for 1 min. Sample was then cooled at a rate of 10 °C/min from 250 °C to −40 °C, followed by heating at 10 °C per min from −40 °C to 250 °C. The degree of crystallinity *Xc* (%), temperatures (glass transition (*T_g_*), crystallization (*T_c_*), and melting (*T_m_*), and enthalpy changes (melting (Δ*H_m_*) and crystallization (*H_c_*) were calculated from thermograms as shown in Table 1. The degree of crystallinity (*Xc*) of PLLA in the nanocomposites was calculated by the following formula Equation (2).
(2)$$Xc\left(\%\right)=\frac{\Delta H_{m}-\Delta Hc}{wt\%\times \Delta H_{m}{}^{\circ}}$$
where Δ*H_m_*° = 93.7 J/g when PLLA is considered as 100% crystalline.

Tensile characteristics were evaluated using a Shimadzu Corporation, Long Beach, CA, USA TRAPEZIUM-X Universal Testing Machine (AG-20KNXD Plus) with a crosshead speed of 1 mm/min. Using a hand cutter, the nanocomposites were cut with dimensions of 10 mm × 50 mm (width × length) and a gauge length of 20 mm (ASTM D6287) was maintained during these experiments.

Swelling tests with nanocomposites were performed as follows: 10 mm × 10 mm nanocomposites with known weight were placed in preweighed Eppendorf tubes filled with 1 mL of PBS. Eppendorf’s were shaken at 37 °C in a shaker. The nanocomposites were gently dried by tissue and weighed for the increase in weight by PBS at various periods. For each time point, all nanocomposites were examined for 30 days after which no change in weight was observed. The residual weight % was computed and plotted with respect to time.

Change in weight because of collagenase type II enzyme activity, at a fixed temperature (37 °C human body temperature), was noted to evaluate the in vitro degradation test of nanocomposites. The sample size was 10 mm × 10 mm. Before weight measurements at various time intervals, materials were freeze dried overnight to remove any moisture. The remaining weight percent was determined. For each measurement, triplicate nanocomposites samples were taken. Nanocomposites measuring 10 mm × 10 mm were placed in Eppendorf’s with a cell culture medium to check the pH changes at day 1, day 3 and day 7. The influence of metabolic products on cell culture media was studied by observing their pH values. Triplicate samples were used for this test.

Antibacterial test. An antibacterial test of the PLLA/nHA and PLLA/COC10-nHA was carried out using zone inhibition (ZI) assay. Four bacterial strains, namely *Escherichia coli* (*E. coli*), *Staphylococcus aureus* (*S. aureus*), *Pseudomonas aeruginosa* (*P. aeruginosa*), and *Listeria monocytogenes* (*L. monocytogenes*), were used. This method was carried out on solid agar petri dishes. Nanocomposites of 6 mm diameter-sized discs were cut, sterilized under UV for 15 min, then placed on the *E. coli*, *S. aureus*, *P. aeruginosa* and *L. monocytogenes* inoculated agar plates. Plates were incubated for 24 h at 37 °C, then the zone of inhibition was recorded. Triplicate nanocomposites were used for analysis.

Cell culture. To test the biocompatibility of the nanocomposites, frozen cell lines were passaged five times in complete media (DMEM + 10% FBS + 1% pen-strep) before being incubated at 37 °C in humidified air with 5% CO_2_. After, 70% confluency cells were passaged, and media was replaced every 48 h. All the nanocomposites were cut into 15 mm diameter, spherical, disc-shaped scaffolds. Before in vitro studies, nanocomposites were sterilized by soaking them in 70% ethanol for 2 h, then washing them in PBS. A 24-well tissue culture plate was used to press the nanocomposites in its wells. BMSC and MC3T3-E1 at a concentration of 1 × 10^4^ cells per well were sown onto the sample discs. Both cell lines were cultivated for 1, 14, and 21 days in complete media. After days 1, 14, and 21, the culture media was withdrawn. After that, the cells were treated for 2 h at 37 °C with PrestoBlue reagent (10% in only minimal essential media). The incubated cell viability reagent was then transferred to a 96-well plate in a volume of 100 μL. The cell viability of the cells was determined by a fluorescence plate reader at 535–560/590–615 nm fluorescence intensity excitation/emission. For the analysis, in three independent tests, triplicate nanocomposites were used. Data was provided as an average standard deviation (SD), and a difference of <0.05 was used. To compare all pairings of columns, the data were subjected to a one-way Analysis of variance (ANOVA) analysis using an appropriate Tukey procedure for statistical comparison.

Live/dead assay was performed using LIVE/DEAD^®^ Viability/Cytotoxicity Kit (mammalian cells) on MC3T3-E1 and BMSC at 1, 7, and 21 days. The cells were cultivated using the same procedure as before. As a control, cells that had not been treated were used. Freshly prepared calcein AM (stains live cells green because of intracellular activity) and ethidium homodimer-1 (stains dead cells red because of loss of cell membrane) together in 50 μL were added to a 24-well plate after washing them with PBS on days 1, 7, and 21. Cells attached to the nanocomposites were incubated for 30 to 40 min before being observed using an inverted fluorescence microscope (ZEISS Axio Observer Z1, Zeiss, Jena, Germany) at 494/517 nm and 528/617 nm for calcein AM for ethidium homodimer. Living cells were counted from three independent replicates of nanocomposites using the Image J program. The percentage of living cells was calculated by dividing the number of live cells by the total number of cells. At 10X magnification, four random field photos were captured.

F-actin/DAPI staining was used to examine cell adherence and proliferation at 2 weeks (14 days) and 3 weeks (21 days) after cell seeding on samples. After removing the medium, the cells were rinsed with PBS before being fixed with 4% PFA for 15 min, then permeabilized with 0.1% Triton X-100 solution for 30 min. Then, for 1 h, 1% BSA was added. The wells were filled with F-actin/DAPI in PBS. An inverted fluorescent microscope was used to examine the cells (ZEISS Axio Observer Z1, Zeiss, Jena, Germany). From duplicate samples, four random photos were collected.

Mineralization studies. The mineralization was identified by using an Alizarin Red test. MC3T3-E1 and BMSC were planted in appropriate numbers on the nanocomposites and grown for 2 weeks (14 days) at standard incubating conditions. After removing the media, the cells were immobilized by using PFA (4%) for 10 min. For 45 min, nanocomposites were stained with 1% Alizarin Red prepared in phosphate buffer solution, then rinsed in PBS to eliminate nonspecific staining. The nanocomposites were examined using a fluorescence microscope (ZEISS Axio Observer Z1, Zeiss, Jena, Germany). From duplicate samples, four random photos were collected.

## 3. Results

### 3.1. Morphology and Structure

#### 3.1.1. Morphology of the Nanocomposites

Size and morphology of lab-synthesized nHA were evaluated by SEM as shown in Figure 1a–e. The size of nanoparticles was found to be in the range of 20–35 nm, while the shape was found to be spherical. To visualize the interaction between the polymer and the nanoparticles and morphology of the nanocomposites, cryofractured surface SEM images were taken as shown in Figure 1f–o. Figure 1f,k shows the polymer matrix of PLLA and PLLA/COC10. PLLA film had a compact and smooth structure because PLLA does not contain many side chains or bulky groups that could interfere with the packaging of the polymeric chains. In PLLA/nHA nanocomposites, when the nHA wt% was low, there was a clear formation of the porous structure in nanocomposites; when nHA wt% was higher, pores were fewer and smaller, while some pit formation was observed in nanocomposites. During the cryofracturing, breaking points of the film were the points of PLLA and nanoparticle interface. In the case of PLLA/nHA, at different concentrations, the nHA was distributed randomly and homogenously; in some areas, nHA were embedded in the pits and some were distributed in plain areas. The PLLA/COC10 film surface was not smooth as compared to pristine PLLA, because here, COC had bulky molecules that were disturbing the PLLA chain packing. In the case of PLLA/COC10-nHA at lower wt% of nHA, there was a homogeneous distribution of nHA, while moving towards higher wt% of nHA, such as 20 wt% of nHA, the agglomerates could be easily seen. We believe that when the ratio of the reinforcing agents to the polymer exceeded a critical value, hydrogen bonding between PLLA and nanoparticles would cease to exist at higher wt% of nHA, resulting in the agglomeration. A similar phenomenon was observed in the case of PLLA/MCC/HA composites [29]. Agglomeration affected the mechanical properties and resulted in increased brittleness of the nanocomposites.

#### 3.1.2. Wide-Angle XRD of Nanocomposites

XRD was carried out for the phase confirmation of the synthesized nHA. The XRD pattern of nHA shown in Figure 2a has characteristic diffraction peaks of HA at 2*θ* = 25.8°, 31.9°, 32.9°, 34.0°, and 39.8°, corresponding to crystal planes of HA (002), (211), (300), (202), and (310), respectively. XRD data matched the JCPDS 09-0432 and was in agreement with the already reported data [30]. The crystallite size of synthesized nanoparticles by using the Scherer formula was found to be 22.3 nm.

XRD patterns of both PLLA and PLLA/COC10 had PLLA α-phase (unit cell dimensions a = 1.06 nm, b = 0.106 nm, and c = 2.88 nm). PLLA α-phase reflects intense peaks at planes (110)/(200) at angle 16.7° and (203)/(113) at angle 19.0° and other planes (011) and (211) reflect at angle 14.6° and 22.3° at angle as shown in Figure 2b [11,31]. On addition of nHA (1–10 wt%) in PLLA/COC10 composites, diffraction peaks of PLLA showed increased intensity as compared to the neat PLLA. These results indicated PLLA/COC10-nHA had increased crystallinity as compared to PLLA because of the nucleation effect of nHA and COC for PLLA chains. On further increasing nHA content to 20 wt%, the characteristic peak intensity of PLLA started decreasing.

Observation of the position of the XRD pattern indicated that the diffraction peaks appeared at intermediate values assigned in the literature to the pure α and α′ forms of PLLA shown in Figure 2c,d [32]. This revealed an interesting fact about PLLA crystal structure that, on nHA addition, as a tertiary phase in PLLA/COC10 and as a secondary phase in PLLA, the α′ crystal structure started to appear, which existed as a pseudohexagonal crystalline structure with loose packaging of polymeric chains (PLLA) and slightly bigger unit cell parameters, indicated by the broadening of the peaks and shifting towards lower 2*θ* 16.4° and 18.7°. This phenomenon was supported by the reported work of MC Righetti et al. [31] and IGI Athanasoulia et al. [33]. Notably, there was no indication of the immiscibility of the phases from the XRD results. Further, the integral intensity of hydroxyapatite peaks at 25.8° and 34.0° corresponding to the (002) and (211) reflections increased on increasing nHA concentrations in PLLA and PLLA/COC10 nanocomposites. At lower concentrations, 1 wt% to 10 wt% nHA, strong and sharp intensity of the peak (110)/(200) indicated that nHA was acting as a nucleating agent for PLLA chains and dispersed evenly. nHA was causing PLLA/COC-nHA nanocomposites crystallinity to increase by 114% greater than the PLLA/COC10 blend and 128% higher than PLLA at 10 wt%. This could be justified by the fact that both COC and HA were acting as nucleating agents and imparting crystallinity to PLLA/COC-nHA. At 20 wt% incorporation, all nanocomposites exhibited considerable decrease in the crystallinity [34].

#### 3.1.3. FTIR Analysis

FTIR was carried out to check the interaction between inorganic filler and organic polymer matrix in nanocomposites. No chemical degradation of any constituent nanocomposites was indicated in FTIR spectra. FTIR was used to identify functional groups present in the nanocomposite, as well as to analyze the chemical change between nanocomposites after mixing. The synthesized nHA compound was subjected to FTIR as shown in Figure 3a. The presence of peaks at 1092 (PO_4_^3−^, ν_1_), 1023 (PO_4_^3−^, ν_1_), 963 (OH^−1^), 601 (HPO_4_^2−^), and 561 cm^−1^ (PO_4_^3−^, ν_3_) were the characteristic bands for nHA confirming its successful synthesis [35,36]. A very small peak of OH was visible at 3571 cm^−1^ affirming that the amount of H_2_O was very low. Particularly, a distinguishable band at 963 cm^−1^ was observed, corresponding to an asymmetric P–O stretching vibration of PO_4_^3−^. In addition, a medium sharp peak appearing at 631 cm^−1^ corresponded to the O-H deformation mode.

Figure 3d displays the FTIR spectra of PLLA/nHA nanocomposites. PLLA had characteristic sharp peaks at 1753 cm^−1^ and 1178 cm^−1^ that corresponded to C=O, and C–O stretching peaks appearing at 1044 cm^−1^ and 868 cm^−1^ represented the C–CH_3_ bending and C–O–C stretching of PLLA, respectively as shown in Figure 3b. IR peaks of both PLLA and nHA appeared for PLLA/nHA nanocomposites, and their peak positions were identical to individual constituents, confirming the absence of chemical bonding between PLLA and nHA. However, a broadband at 1023 cm^−1^ was observed for nanocomposites containing 5 wt% of nHA and higher, due to physical interaction between CH_3_ of PLLA and PO_4_^3−^ of nHA. The broadening of this band could be attributed to the formation of hydrogen bonding. Peak intensity at 1753 cm^−1^ of C=O (PLLA) decreased, while intensity at 1023, 629, 601, and 559 cm^−1^ increased, proportional to the concentration of nHA added in nanocomposites, revealing that nHA had uniformly blended with PLLA [37].

FTIR spectra of the second set of nanocomposites PLLA/COC10 with nHA are shown in Figure 3c. Three major areas of interest were in the fingerprint region, i.e., 1. C=O region (1800–1700), 2. CH_3_ and CH bending area (1500–1300), and 3. C–O–C (1300−950). The band intensity at 1212 cm^−1^ and 1185 cm^−1^ corresponded to the C–O–CH_3_ stretching mode of crystalline and the C–O–C ester group of the amorphous part of PLLA, respectively. The decrease in intensity of peaks indicated the changes in the crystalline structure of PLLA after mixing with COC. Further, adding nHA to PLLA/COC10 blends, the characteristic peak of PO_4_^3−^ at 963 cm^−1^ appeared and merged with the PLLA peak at 1021 cm^−1^, including other peaks at 629, 600, and 550 cm^−1^. In PLLA/COC10, the C–O–C peak was at 878 cm^−1^, while in PLLA/COC10-nHA, the C–O–C peak shifted to 866 cm^−1^. Similarly, the C–CH_3_ peak shifted from 1082 cm^−1^ to 1088 cm^−1^ in the PLLA/COC10-nHA nanocomposite. In addition, a peak at 921 cm^−1^, characteristic of 10^3^ helix formation [38] of PLLA, appeared as a shoulder and intensified with the increase in percentage of nHA. Thus, nHA incorporation in the PLLA/COC blend resulted in increasing PLLA crystallinity. Interestingly, the peak at 1046 cm^−1^ moved to 1026 cm^−1^, which was attributed to the hydrogen bonding [39] between CH_3_ of PLLA and PO_4_^3−^ of nHA, and the peak broadened with increased percentage of nHA.

### 3.2. Thermomechanical Properties

#### 3.2.1. DSC of Nanocomposites

DSC thermograms help to understand more about the structure and crystallinity of the nanocomposites. Figure 4 shows the DSC thermograms for the polymeric nanocomposites heated, cooled (Figure 4c,d), and again heated (Figure 4a,b) at 10 °C per minute. Using these thermograms, values of glass transition temperature (*T_g_*), cold crystallization temperature (*T_cc_*), and melting points (*T_m_*^1^ and *T_m_*^2^) were calculated and are presented in Table 1. In general, DSC thermograms showed that *T_g_* was shifted to lower values after the addition of nHA.

In cooling curves of PLLA and PLLA/COC10, Δ*H_c_* and *T_c_* values of PLLA/COC10 were lower than pristine PLLA. No exothermic crystallization curve was observed during the cooling cycle of nanocomposites (Figure 4c,d,f); the only thermal transition was observed around *T_g_*. Data obtained from second heating curves showed that *T_g_*, *T_cc_*, *T_m_*^1^, and *T_m_*^2^ shifted to lower values with the addition of the nHA as compared to pure PLLA [40] and PLLA/COC10 (Figure 4a,b).

In the second heating curve of PLLA/nHA nanocomposites, *T_g_* values decreased from 60 °C for pure PLLA to 45 °C for 1 wt% nHA, 46 °C for 5 wt% nHA, 46 °C for 10 wt% nHA, and 44 °C for 20 wt% nHA. There was no *T_cc_* observed before *T_m_* for pristine PLLA, while all PLLA/nHA had distinguished *T_cc_*, which is depicted in Table 1 and Figure 4b. This exothermic peak was due to the rearrangement of molecular chains in the amorphous domain. Pure PLLA had one *T_m_* at 176 °C, while HA nanocomposites had two melting peaks at lower temperatures than pure PLLA.

For PLLA/COC-nHA nanocomposites, all four peaks of *T_g_*, *T_cc_* (between 108 to 97 °C), *T_m_*^1^, and *T_m_*^2^ appeared in the second heating scan. *T_g_* values decreased from the PC-10 blend at 62 °C by 5–7 °C in PLLA/COC-nHA nanocomposites. PLLA/COC-nHA 1 wt% had *T_g_* at 43.7 °C, which increased to 45.7 °C in the case of the PLLA/COC-nHA 20 wt%. *T_cc_* values appeared as a broad cold crystallization endothermic peak at 103 °C just before two melting peaks in all PLLA/COC-nHA. Two melting peaks (around 150 °C) appeared much below the melting peak (*T_m_*) of the pure PLLA. In the literature, He et al. and Athanasoulia et al. [38,40] reported and associated these double-melting peaks of PLLA with simultaneous melt crystallization, i.e., two lamella thicknesses in which the smaller one appeared at a lower temperature, while the larger one appeared at higher temperatures [41]. Several studies suggested that melt recrystallization was the dominant mechanism. This included the melting of original PLLA crystals at a lower temperature, their further rearrangements into perfect crystallites, followed by a final melting of recrystallized and perfected crystallites [32]. Reduction in the melting temperature was a trend followed by many crystalline polymers forming miscible blends with other polymers. *T_g_* values of polymer decreased in nanocomposites indicated the good dispersion of the nHA. nHA was acting as the nuclei for the formation and crystal growth of polymer. Nanoparticles seemed to increase the molecular mobility of the polymeric chains at lower temperatures, while decreasing the free volume of the polymeric chain at the same time. Another interesting fact was observed that the presence of the exothermic peak before the melting peak indicated the presence of the conformationally distorted α′ phase converted to the thermodynamically favorable α phase [40], as corroborated by the XRD measurements. From Δ*H_m_* and Δ*H_c_*, percentage crystallinity of the polymer systems was calculated as shown in Table 1. The decrease in crystallization enthalpy Δ*H_c_* of PLLA in nanocomposites as compared to the pure PLLA was affected because of two mechanisms: the first was the nucleation effect of HA, and the second was the hindrance of PLLA chain mobility because of nanoparticles present in the matrix. Intriguingly, this mechanism increased in the case of PLLA/COC10, because here, in addition to HA, the COC chains were not only causing nucleation but also hindering PLLA chain mobility [34]. At lower nHA concentrations, nanocomposites showed an increase in crystallinity, showing maximum values at 1 wt% nHA because of the nucleation effect leading to the hindrance effect. In 20 wt% nHA, the agglomeration affected the interaction between polymer and nanoparticles. Here, nucleation was disturbed and hindrance of PLLA chains was increased, thus the crystallinity was decreasing.

#### 3.2.2. Mechanical Properties of Nanocomposites

Figure 5a shows stress-strain curves obtained from tensile mechanical testing of PLLA, PLLA/COC10, and different loadings of nHA in PLLA/COC10-nHA and PLLA/nHA nanocomposite films, respectively. Different parameters obtained from stress-strain curves were detailed in Table 2 (young modulus), Table 3 (strain (%) at break), and Table 4 (maximum stress). PLLA is a rigid polymer [19,41] having high tensile strength but ruptures at the elongation at break value of 7.84% strain. PLLA/COC10-nHA showed improved tensile mechanical properties as compared to PLLA/nHA film. Ultimate tensile strength (UTS) (as shown in Figure 5d) and strain percent values at elongation at break (as shown in Figure 5b) were dependent on the polymer and nHA wt% in the nanocomposite. Increasing the nHA wt% resulted in both decreasing UTS and elongation at the breakpoint. PLLA/PVDF blends showed improved ductility in the presence of compatibilizers [42]. COC in PLLA/COC10-nHA resulted in strong and ductile polymer; thus, its nanocomposite with nHA showed elongation at breakpoint at higher strain percent values than PLLA/nHA. PLLA/nHA showed maximum tensile strength at 1 wt% nHA. PLLA/COC10-nHA that showed maximum strain percent values of 17.14 ± 2.6 at 1 wt% nHA loading appeared tougher nanocomposite materials than all other nanocomposites of interest. Ultimate strain percent values of 1 wt% nHA loading in PLLA were 12.74 ± 1.34 and 1 wt% nHA loading in PLLA/COC10 were 10.29 ± 0.9. Comparison of strain % or elongation at break values at 1 wt% nHA loading showed 155% and 215% increase, respectively, as compared to pure PLLA films. Upon further increasing nHA content in the films to 5 wt%, then 10 wt%, the ultimate strain % decreased but was still higher than pristine PLLA. Elongation at break strain % for PLLA/nHA 5 wt% was 10.37 ± 1.2 MPa, as compared to PLLA/COC10-nHA 5 wt%, which had 14.2 ± 0.98 MPa, while in PLLA/nHA 10 wt% elongation at break values was 5.56 ± 2.1 MPa lower than elongation at break of PLLA/COC10-nHA 10 wt% at 12.71 ± 1.33 MPa. UTS values of PLLA/nHA were higher than the PLLA/COC10-nHA but lower than pure PLLA and PLLA/COC10. At lower nHA loading, this decrease in UTS was 10%, which increased to 42.5% in the case of 10 wt% nHA. One possible reason might be that at lower content of nHA, there was hydrogen bonding between the PLLA and nHA as corroborated by FTIR analysis. The reinforcing effect of the nHA increased the elongation at break values of the polymer nanoparticle system. Decrease in UTS might be because of nHA agglomeration in the polymer matrix decreasing the reinforcing effect, leading to failure of the polymer at lower tensile strength values. In the case of PLLA/COC10-nHA, decrease in UTS was because of the three-phase interaction among PLLA/COC and nHA that resulted in non-alignment of the polymer chains, leading to a decrease in UTS.

However, interesting results were revealed by comparing the young’s modulus values. Young’s modulus values of the pure PLLA was 571.6 MPa, while PLLA/COC10 was found to be 870 MPa. In nHA-impregnated PLLA films with increasing nHA content, there was an increasing trend of young’s modulus (as shown in Figure 5c). For 1, 5, and 10% nHA in PLLA/COC10-nHA, young’s modulus values were 104, 271.5, and 280.5 MPa. Values of Young’s modulus in PLLA/nHA were lowered by 16.5%, 62%, and 62.8% in 1 wt%, 5 wt%, and 10 wt% PLLA/nHA, as compared to PLLA/COC10-nHA, because of agglomeration. These results showed that PLLA/COC10-nHA had enhanced mechanical properties as compared to the PLLA/nHA. These results lead to the assumption that COC chains were not only reinforcing the polymer matrix, but were also involved in the better dispersion of nHA in polymer matrix as corroborated by SEM [43].

Cancellous bone requires a Young’s elastic modulus of 0.1–4.5 GPa, while that of cortical bone is 17 GPa [44]. Depending upon these values of mechanical testing, above mentioned films were found to be a good candidate for the cancellous bone-tissue engineering and dressing application. Specifically, these nanocomposites can be used in mechanically active areas. Results revealed that the mechanical properties of the polymers can be fine-tuned by changing the nanoparticle concentration in the nanocomposites.

### 3.3. Swelling Analysis of Nanocomposites

Analysis of the swelling ratio of nanocomposite material is an essential parameter to understand the solute diffusion, surface, and mass transfer properties of the composites. Additionally, the swelling behavior of nanocomposites in water mimics the environment of the body. Therefore, to understand the swelling properties, nanocomposites PLLA/nHA and PLLA/COC-nHA were placed in PBS buffer, and swelling properties were observed over time. Figure 6a shows the swelling profiles of nanocomposites at different time intervals from 1 day to 30 days. PLLA had the lowest swelling ratio compared to all composites [45,46]. The swelling ratio of PLLA/nHA had increased to two- to three-fold as compared to PLLA. The increase in swelling behavior of PLLA/nHA contributed to an accumulation of water molecules by interacting with hydroxyl groups present in nHA through hydrogen bonding (as shown in FTIR). Intriguingly, a decrease in swelling ratio was observed with increasing the concentration of nHA from 1 wt% to 20 wt%. At a lower concentration of nHA (1 wt%), the swelling ratio was 29.9%, which was greater than 20.9% at 20 wt% nHA for PLLA/nHA. It can be seen from SEM images that nHA nanoparticles tended to aggregate upon increasing the concentration to 20 wt% creating dense network structure in polymeric chains. As a result, nanoparticles showed less interaction with water to retain within the PLLA/nHA [47]. Thus, a decreasing trend in the swelling ratio of PLLA/nHA was observed. Similarly, the addition of HA from 1 wt% to 20 wt% to nanocomposites PLLA/COC-nHA exhibited a decrease in swelling ratio. The swelling ratio of PLLA/COC10-nHA 1 wt% was around 23.2% and was reduced to 13.4% for PLLA/COC-nHA 20 wt%. Both PLLA/nHA and PLLA/COC-nHA nanocomposites showed maximum swelling behavior during the initial 24 h, and they reached equilibrium after 24 days. It was also noted that the swelling ratio of PLLA increased from 8.6% to 23% by the addition of 10 wt% COC in PLLA/COC10. The enhancement of swelling was attributed to the amorphous nature of COC copolymer for accommodating water molecules due to the disordered arrangement of COC polymeric network. Therefore, the additional nHA could fine tune swelling properties of the nanocomposites for bone-tissue engineering [48].

### 3.4. Degradation Analysis

Degradation kinetics of the nanocomposites provided necessary information about the rate at which nanocomposites disappeared in the presence of chemical factors that were present in the extracellular matrix. To investigate degradation characteristics of blends and nanocomposites, the various compositions of PLLA/nHA and PLLA/COC-nHA hybrid composites were incubated in collagenase solution over time in a humidified incubator. Figure 6b represents the degradation kinetics of blends and nanocomposites in enzymatic conditions which is more useful than hydrolytic degradation [49,50]. During incubation, the degradation rate of PLLA and PLLA/COC10 nanocomposites were increased over time and the rate of degradation was higher for PLLA alone. The rate of degradation for PLLA was observed at 93% after 7 days of incubation, reaching 30% after 48 days [51,52]. Moreover, upon increasing the concentration of HA, the degradation rate of PLLA/nHA was decreased. The maximum degradation rate was observed for nanocomposites containing 1 wt% nHA, 80.4% after 7 days, reduced to 2% after 48 days. Intriguingly, a similar trend was observed for nanocomposites containing PLLA/COC-nHA. In the case of PLLA/COC-nHA, a maximum degradation rate was observed for 1 wt% nHA, with 83.4% after 7 days, reduced to 3% after 48 days. Although PLLA/nHA and PLLA/COC10-nHA exhibited the same trend of degradation upon addition of nHA, PLLA/COC10-nHA showed a slower degradation profile compared to PLLA/nHA. These results indicated the importance of COC polymer in regulating the degradation of nanocomposites. The decrease in degradation of the polymer upon addition of nHA was attributed to aggregation of nHA nanoparticles in composites, thereby creating PLLA/COC10-nHA material denser than PLLA/nHA to interact with chemical factors [53]. Thus, the addition of nHA improved the strength of composites. However, no significant difference in degradation was observed for PLLA and PLLA/COC10. Addition of nHA and incorporation of COC in PLLA emphasized the importance in regulating the degradation rate of composites.

PLLA/nHA and PLLA/COC10-nHA were immersed in the media along with the cells, and pH changes were observed on day 1, day 3, and day 7 (Figure 6c). pH measurements revealed that the solution alkalinity (high pH) was because of the release of HA into the solution. While the reduction in pH was because of the polymer’s acidic by-products, S. Hassanajili suggested that the nHA acted as a buffer and reduced the swelling effects caused by acidic by products of the polymer degradation [52].

### 3.5. In Vitro Studies of Nanocomposites

#### 3.5.1. Antimicrobial Activity of Nanocomposites

Antimicrobial activity of compounds was tested by the Kirby-Bauer test, also known as the zone of inhibition (ZI). The antimicrobial activity of PLLA, COC, PLLA/nHA, and PLLA/COC/nHA with different nHA concentrations from 1 to 20 wt% were evaluated against the four most common bacteria found in the environment. *E. coli* and *P. aeruginosa* were Gram negative, while *S. aureus* and *L. monocytogenes* were Gram positive bacteria. Antibacterial activity is shown in Appendix A. Gentamycin was used as the positive control, while the polymer system acted as a negative control. No inhibition zone was found for PLLA, which revealed it does not have any antibacterial potential [53,54], or PLLA/COC10, indicating no antibacterial potential of the two polymers. PLLA/nHA at all concentrations did not exhibit significant antibacterial activity with the four bacteria, owing to the very compact packing of the nanoparticles inside the PLLA matrix as indicated by SEM. Surprisingly, PLLA/COC10-nHA displayed a significant response to bacteria at higher nHA loading, except with *E. coli*, a Gram-negative bacterium [55]. The diameter of zone of inhibition or no bacterial growth was calculated by vernier calipers, and the values were tabulated as shown in Table 5. PLLA/COC10-nHA 5 wt% showed very small ZI for *S. aureus*. Based on data in Table 5, PLLA/COC10-nHA 10 wt% exhibited the highest potency for bacterial inhibition: 15.34 mm, 14.73 mm, and 10.39 mm against *S. aureus*, *L. monocytogenes*, and *P. aeruginosa*, respectively. PLLA/COC10-nHA 20 wt% showed high antibacterial potential against *L. monocytogenes* (ZI = 12.68 mm) and *P. aeruginosa* (ZI = 13.29 mm), respectively, while for *S. aureus*, ZI was found to be 8.21 mm. These results showed that PLLA/COC10-nHA had in vitro activity against a broad range of Gram-positive and Gram-negative bacteria with high activity against Gram-positive bacteria [56]. The reason for the largest zone of inhibition of Gram-positive bacteria strain could be attributed to the absence of an outer layer membrane on bacteria cell walls, whereas the presence of a thick outer layer of the lipid membrane and lipopolysaccharides protect the Gram-negative bacteria from antimicrobial compounds [18,57]. Intriguingly, *P. aeruginosa* Gram-negative bacteria showed a significant response to PLLA/COC10-nHA. Reportedly, COC is a hydrophobic polymer and has low permeability of oxygen and moisture, and the films made by COC could behave as a barrier between bacteria and the surrounding environment, thereby preventing nutrients to reach bacteria. As a result, bacterial growth was reduced due to oxidative stress [57]. Moreover, the antibacterial activity of blend were increased with increasing concentrations of nHA in *P. aeruginosa* [58]. The reduction of oxygen permeability was decreased with increasing concentrations of nHA due to the longer diffusive path that the oxygen must travel in the presence of nHA nanoparticles [55].

#### 3.5.2. Cell Culture on Nanocomposites

MC3T3-E1 and BMSC attachment, proliferation, viability, and mineralization studies on the surface of the PLLA/COC10-nHA and PLLA/nHA nanocomposites are shown in Figure 7. Cell fluorescent intensity and viability for day 1, day 7, and day 21 for the two cell lines are shown in Figure 7g,h by using PrestoBlue assay. Intriguingly, for the first couple of days, nanocomposites showed almost no difference, with cells attaching to the surface of the nanocomposites. However, after that, PLLA/COC10-based nanocomposites showed more than 91% cell viability on day 14 (Figure 7e,f), while this value enormously raised to 96% by day 21. The cell viability data are highly positive, indicating that no cell death occurred. According to the ISO 10993-5:2009 standard, there was no compromise on biocompatibility based on these cytocompatibility results [48]. Because of acceptable sample roughness and surface features, cell concentration increased with increasing nHA concentration up to 10 wt% (Figure 7a,b). This showed that nHA reinforcement in PLLA and PLLA/COC10 scaffolds increased osteogenesis [59,60]. However, reaching up to 20 wt%, the nHA concentration adversely affected the cell viability and proliferation (Figure 7c,d). Osteoblast cells proliferated dramatically on the composites having nHA, as compared to the only polymer systems during 21 days of culture (Figure 8). It was vivid from the cell orientation and development pattern that, when nHA concentrations were added, nHA provided a site of attachment for the cells. Moreover, the cells also showed higher compatibility and growth to the nanocomposites with higher mechanical properties. In a previous study, PLLA/COC10-blend systems [11] demonstrated potential as a biocompatible material with favorable interactions with cells. Here, the osteoconduction was introduced by the incorporation of the nHA in PLLA/COC10 blends.

Noticeably, Figure 8 and Figure 9 show that cell death was very low in the LIVE/DEAD experiment. The planted cells were well adhered to the nanocomposites on day 1. There was an increasing number of cells attached to the surface of the scaffolds having nHA because the polymer surface was hydrophobic, while the insertion of the HA into these scaffolds increased the hydrophilicity of the system after day 1 cell attachment. It was evident that in both kinds of nanocomposites, increasing the concentration of the nHA from 1 to 10 wt% accelerated cell proliferation of 10- to 30-fold from day 1 to day 21. On days 14 and 21, the LIVE/DEAD assay revealed that the cells had branched and linked networks, as well as showing good morphology, indicating an accelerated proliferation rate [61]. In the comparison of PLLA/COC10-nHA and PLLA, PLLA/COC10-nHA had higher cell viability, adhesion, and proliferation of osteoblast and BMSC on its surface [62]. The ANOVA and estimated results as shown in Figure 9 revealed that the cells were proliferating at a very high rate because of the synergistic combination of PLLA/COC and nHA [63]. From day 14 to day 21, PLLA/COC10-nHA demonstrated better cell compatibility in all nanocomposites, and cells were found well populated on the PLLA/COC10-nHA surface. Cells were sensitive to their surroundings, and though no significant morphological differences were seen in all the nanocomposites when compared to PLLA, the combination of PLLA, COC, and nHA had a favorable effect on cellular function.

F-actin and DAPI staining were used to examine cell morphologies in addition to cell proliferation as illustrated in Figure 10 and Figure 11. The PrestoBlue analysis was supported by F-actin and DAPI image analysis with the passage of time [64].

In comparison to pure PLLA or PLLA/nHA, PLLA/COC10-nHA blends demonstrated a higher number of cells. In PLLA/COC10-nHA at 5 wt%, nHA cell density grew by 1.5 times, while at 10 wt%, cell densities rose three-fold. The number of DAPI positive cells per fixed area was used to compute the cell density. Furthermore, when compared to pure PLLA/nHA, cells in the PLLA/COC10-nHA were anchored with lengthy, well-defined filopodia.

Observed under microscope, actin stress fibers first appeared in PLLA on day 2 and in PLLA/COC10-nHA on day 1. Thus, focal adhesion development was detectable in PLLA/COC10-nHA blends from day 2 onwards, whereas it emerged on day 2 in PLLA/nHA. Persson et al. found similar findings in pre-osteoblast cells using PLLA/HA films [62]. BMSC showed increased cell proliferation on days 14 and 21.

In the instance of BMSC, the cell density of the PLLA/COC10-nHA shown in Figure 7, Figure 9, and Figure 12 represented an approximately four time increase in cell density from day 7 to day 14; similar results were obtained when COC/nHA composites were prepared for bone-tissue engineering [17]. The maximum cell viability values were found in PLLA/COC10-nHA 10 wt%. On day 7, a three-fold increase in cell proliferation for PLLA/COC10-nHA 10 wt% was observed with respect to all other nanocomposite materials. Cell adhesion caused the scaffold and cells to connect, resulting in cell attachment and growth on the scaffold surface. Cell adhesion was mediated and controlled by nanocomposite material roughness, swelling behavior, and mechanical properties. Cytoskeleton filaments were harboring on the surface of the scaffold, demonstrating the scaffold’s biocompatibility and aiding in adhesion and proliferation.

Interactions between cellular protein and nanocomposites’ surface appeared to be good based on morphology, cell size, and growth. Figure 10 and Figure 11 show similar results for F-actin and DAPI. The hydrophilic surface promoted cell attachment, whereas the hydrophobic surface was critical for cell growth. Because of the swelling, nHA inside the polymer system and on the polymer surface aided in cell attachment, whereas polymer system hydrophobicity helped in the proliferation [65]. The ideal combination and a balance in the hydrophobicity and hydrophilicity were important. Therefore, the PLLA/COC10 nanocoomposite had more proliferation rate up to 10 wt% nHA concentration than at 20 wt% nHA. Mechanically strong materials had higher cell viability and proliferation as compared to mechanically weak nanocomposite materials. nHA in the polymer matrix and on the surface provided a place for attachment by altering the surface characteristics of the cells. The blend system governed cellular function as a substituent to growth factors. [66]. According to the literature, cell proliferation was influenced not only by the scaffold’s chemical structure but also by its mechanical qualities, which influenced bone cell differentiation and mineralization [67]. Cells responded to mechanical qualities, such as scaffold stiffness, and adjusted their phenotypic and spreading behaviors to achieve the mechanical requirements of the native tissue [68].

The stiff nanocomposite materials may aid differentiation via the mitogen-activated protein kinase (MAPK) and RhoA/Rho-associated protein kinase (ROCK) pathways, according to several studies. Because hard mineralized bone requires higher modulus values than collagenous bone, scaffold materials with other polyesters, such as polyethylene glycol, having lower modulus values demonstrated no mineralization. While in another report, when cells sensed the compatible stiffness in mechanically strong materials, the cells physiology was appreciable [66]. These findings showed a link among shape, swelling, and degradation rate, mechanical properties, and cell viability in PLLA/COC10-nHA nanocomposites for tissue-engineering applications.

#### 3.5.3. Mineralization Studies on Nanocomposites

The MC3T3-E1 model is a well-known in vitro osteogenesis model. Similarly, mineralization on BMSC can help researchers better understand how PLLA/COC10-nHA can be used in bone-tissue creation. As shown in Figure 12 and Figure 13, calcium deposition on PLLA/COC10-nHA by pre-osteoblast and bone marrow-derived stem cell lines was stained with alizarin red S staining. Calcium nodules were a common sign of mature osteoblasts [69,70]. Mineralization rose exponentially from day 7 to day 14. On day 7, there was no discernible difference; however, the calcium deposition was extremely visible on day 14. When PLLA/COC10-nHA cells were compared to control cells and PLLA, the density of calcium nodules rose considerably as is evident in Figure 12. Image J software was used to analyze the images taken on day 14. From Figure 12 and Figure 13, it was clear that the addition of nHA up to 10 wt% had a positive influence on mineralization. Figure 12c,g comparison showed that the PLLA/COC10 10 wt% had the highest bone mineralization because COC addition had added suitable mechanical properties [26] to materials. On PLLA/nHA, similar mineralization results were reported on day 14 [65]. These findings revealed that PLLA/COC10-nHA aided not only in pre-osteoblast cell adhesion and proliferation, but also cell differentiation and maturation. PLLA/COC10-nHA is a potential candidate as a bone scaffold with osteo-induction capabilities, and that the osteo-induction has been improved.

As a result of the foregoing findings, PLLA/COC10-nHA with varied HA concentrations may be easily manufactured with stable features, such as good morphology and decreased crystallinity. When compared to PLLA/nHA, PLLA/COC10-nHA and also with already reported literature in Table 6, PLLA/COC10-nHA nanocomposites exhibited better mechanical, enhanced swelling, and optimized degradation properties and enhanced in vitro cells’ compatibility and mineralization profile with pre-osteoblast and bone marrow-derived cell lines.

## 4. Conclusions

HA nanoparticles were successfully lab synthesized and characterized by FTIR, PXRD, and SEM. PLLA/COC10-nHA and PLLA/nHA composites were successfully prepared by economical physical blending method, i.e., solvent-casting method. Concentration ranges from 1 to 20 wt% HA were incorporated into PLLA/COC10 and PLLA and two systems were compared. SEM images displayed uniform distribution of nHA on and inside the surface of PLLA/COC10 up to 10 wt%; whereas, at 20 wt%, nHA showed agglomeration. Overall better dispersion of nanoparticles was found in PLLA/COC10 as compared to PLLA. Change in functional group position in FTIR also confirmed that nHA influenced the interfacial bonding between PLLA/COC10, while such phenomenon was nonexistent in case of PLLA/nHA nanocomposites. PXRD and DSC results revealed that the α of PLLA/COC10 started transforming to α′ form with the addition of different concentrations of nHA, and both in nanocomposites both forms co-exist in nanocomposites. Additionally, the maximum crystallinity of PLLA/COC-nHA was found (43.3%) at 10 wt% nHA addition. COC and HA both provided nucleating effects, causing an increase in crystallinity. Crystallinity values of PLLA/nHA nanocomposites showed an increasing trend up to 10 wt% addition of nHA, then showed a decreasing trend at 20 wt%, but the degree of crystallinity values was always lower than pure PLLA, PLLA/COC10, and PLLA/COC10-nHA. Furthermore, mechanical properties of PLLA/COC10-nHA nanocomposites exhibited enhancement in young’s modulus, stiffness, and toughness by 141% to 240% as compared to pure PLLA for nHA concentration 1 to 10 wt% HA. PLLA/COC10-nHA had better degradation, swelling properties, and cytocompatibility as compared to PLLA/nHA at different nHA concentrations. PLLA/COC10-nHA 10 wt% was found most effective against Gram-positive and Gram-negative bacteria such as *Pseudomonas aeruginosa*, *Staphylococcus aureus*, and *Listeria monocytogenes*. PLLA/COC10-nHA nanocomposites showed a significant increase in cell viability and proliferation of osteoblasts MC3T3E-1 and BMSC in comparison to PLLA/nHA, while PLLA/COC10-nHA 10 wt% had the maximum cell viability, proliferation, and mineralization for both cell lines because of its mechanical strength. In short, this study revealed that PLLA/COC10-nHA had better mechanical properties, enhanced antibacterial, and suitable in vitro bone-scaffold properties as compared to PLLA/nHA. Moreover, PLLA/COC10-nHA can be fine-tuned according to the need of the bone regeneration. Among all the tested nanocomposites, PLLA/COC10-nHA10 has shown the best properties; therefore, it can be a potential biopolymer nanocomposite material as a replacement for allografts/autografts for injured bone.

## Figures and Tables

**Figure 1 polymers-13-03865-f001:**
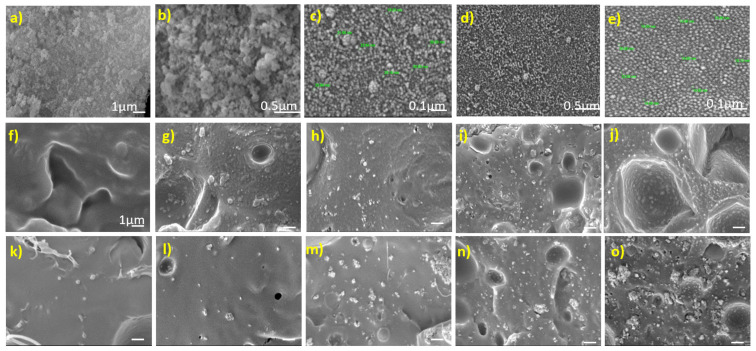
SEM images of pure nHA powder at (**a**) 10,000× magnification, scale bar = 1 µm; (**b**) 40,000× magnification, scale bar = 0.5 µm; and (**c**) 20,000× magnification, scale bar = 0.1 µm. nHA suspended in aqueous solution at (**d**) 0.5 µm and 50,000× and (**e**) 0.1 µm and 100,000×. SEM images of PLLA/nHA nanocomposites (**f**) PLLA, (**g**) PLLA/nHA 1 wt%, (**h**) PLLA/nHA 5 wt%, (**i**) PLLA/nHA 10 wt%, and (**j**) PLLA/nHA 20 wt%. SEM images of PLLA/COC10-nHA nanocomposites (**k**) PLLA/COC10, (**l**) PLLA/COC10-nHA 1 wt%, (**m**) PLLA/COC10-nHA 5 wt%, (**n**) PLLA/COC10-nHA 10 wt%, and (**o**) PLLA/COC10-nHA 20 wt%. Nanocomposites. (**f**–**o**) Scale bar = 1 µm and the images were taken at 10,000 kV.

**Figure 2 polymers-13-03865-f002:**
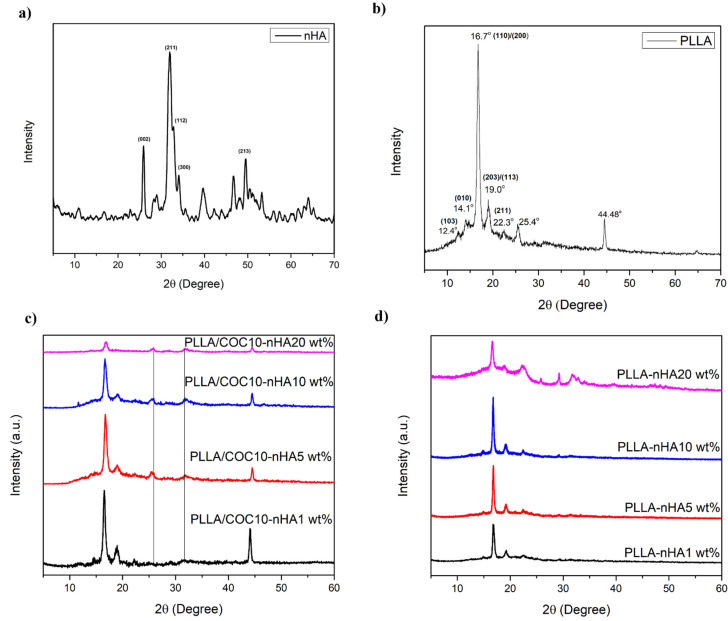
XRD of (**a**) nHA, (**b**) PLLA, (**c**) PLLA/COC10-nHA nanocomposites, and (**d**) PLLA/nHA nanocomposites. Nanocomposites had nHA in 1 wt%, 5 wt%, 10 wt%, and 20 wt%.

**Figure 3 polymers-13-03865-f003:**
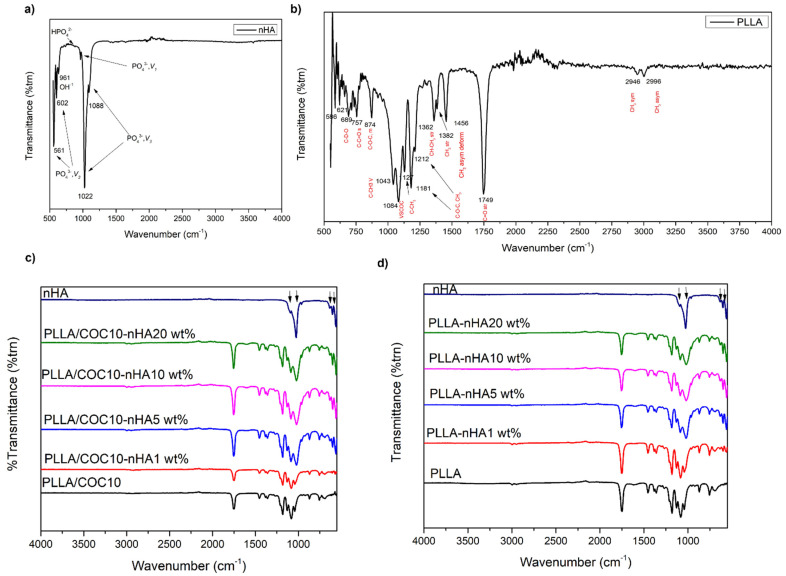
FTIR of (**a**) nHA, (**b**) PLLA, (**c**) PLLA/COC10-nHA nanocomposites, and (**d**) PLLA/nHA nanocomposites. Nanocomposites had nHA in 1 wt%, 5 wt%, 10 wt%, and 20 wt%.

**Figure 4 polymers-13-03865-f004:**
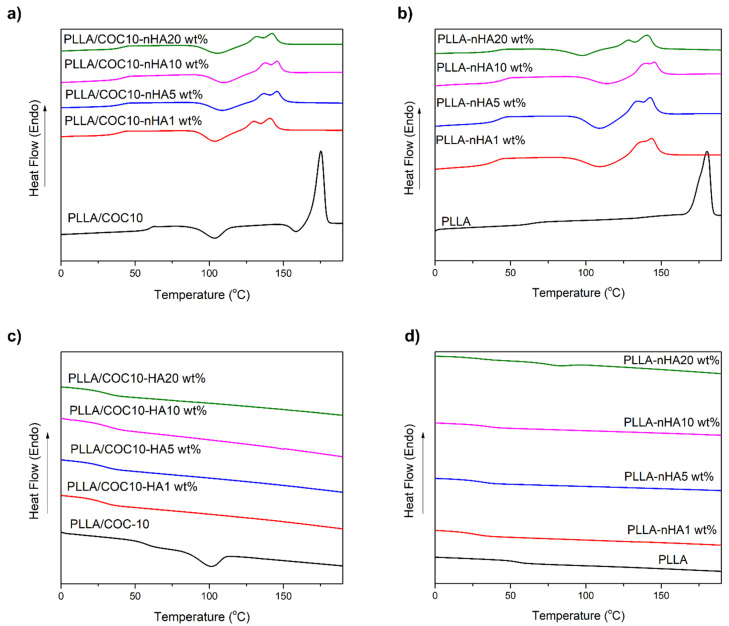
DSC of nanocomposites showed non-isothermal DSC thermograms at 10 °C per minute. PLLA/COC10-nHA (**a**) second heating scan (**c**) non-isothermal melt crystallization. PLLA/nHA (**b**) second heating scan (**d**) non-isothermal melt crystallization scan. Nanocomposites had nHA in 1 wt%, 5 wt%, 10 wt%, and 20 wt%.

**Figure 5 polymers-13-03865-f005:**
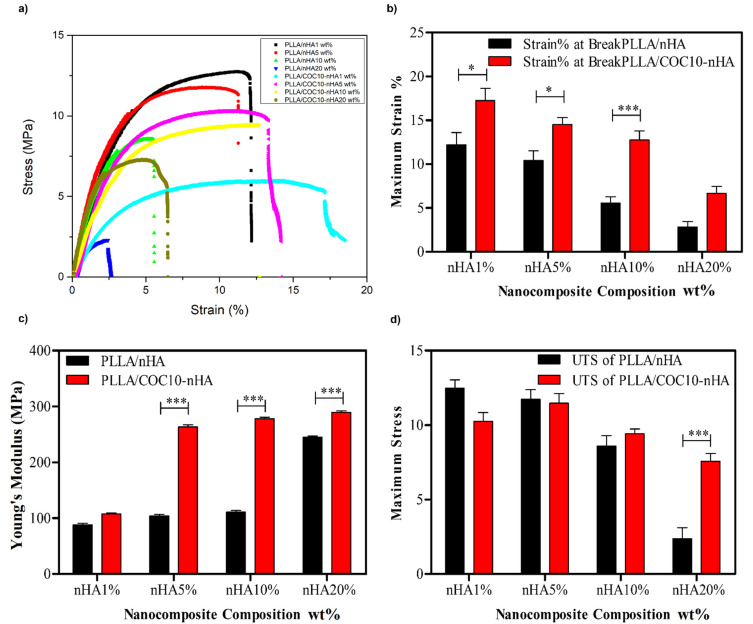
Tensile mechanical properties of the PLLA/COC10-nHA and PLLA/nHA nanocomposites. (**a**) Stress-strain curves of nanocomposites. (**b**) Ultimate tensile strength of nanocomposites. (**c**) Young’s modulus of nanocomposites. (**d**) Elongation at break of nanocomposites. Nanocomposites had nHA in 1 wt%, 5 wt%, 10 wt%, and 20 wt%. Data is statistically significant (error bars: ±SD, * *p* < 0.05, ** *p* < 0.01, and *** *p* < 0.001).

**Figure 6 polymers-13-03865-f006:**
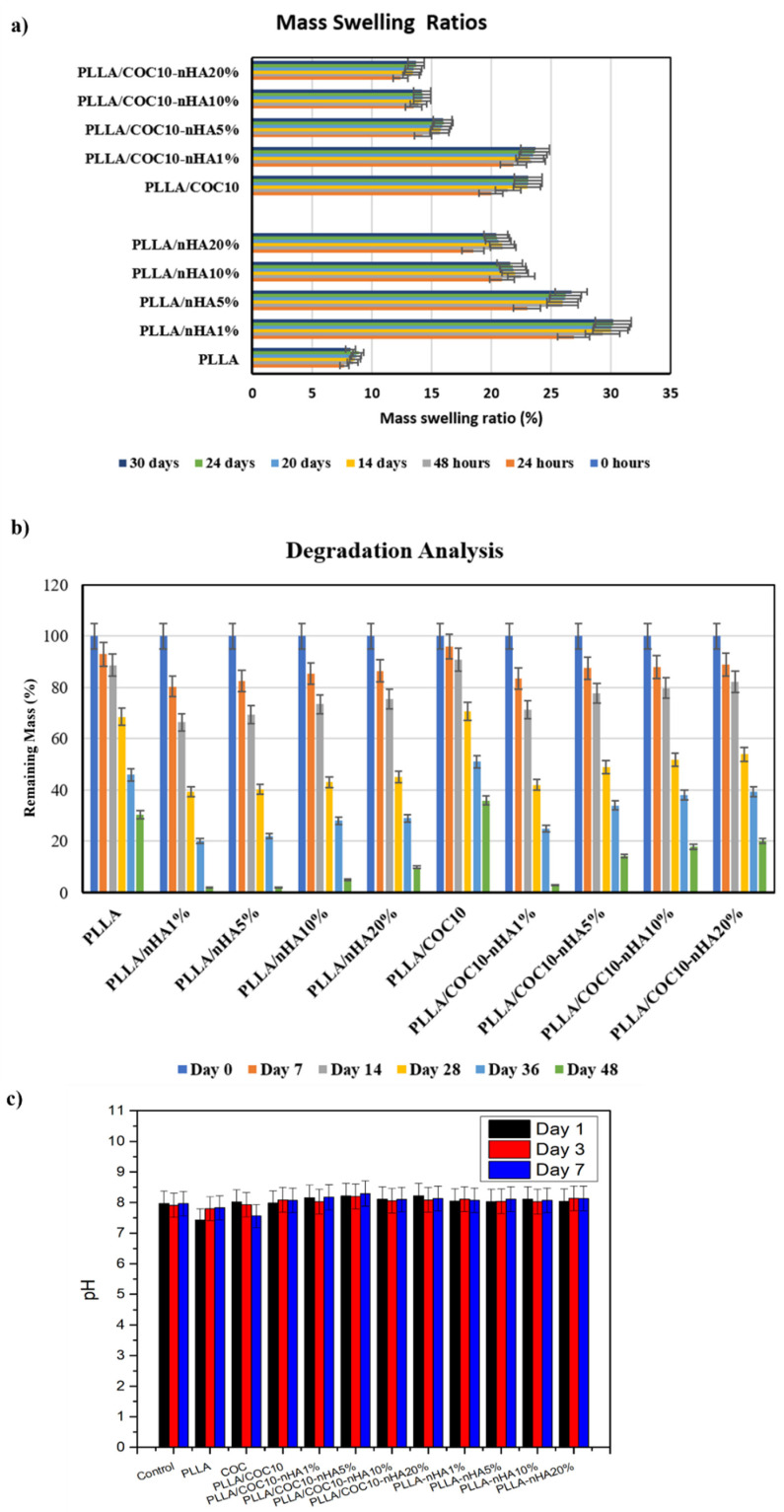
(**a**) Mass swelling ratios of nanocomposites. (**b**) Enzymatic degradation of nanocomposites. (**c**) pH values of nanocomposites with the passage of time. Nanocomposites had nHA in 1 wt%, 5 wt%, 10 wt%, and 20 wt%.

**Figure 7 polymers-13-03865-f007:**
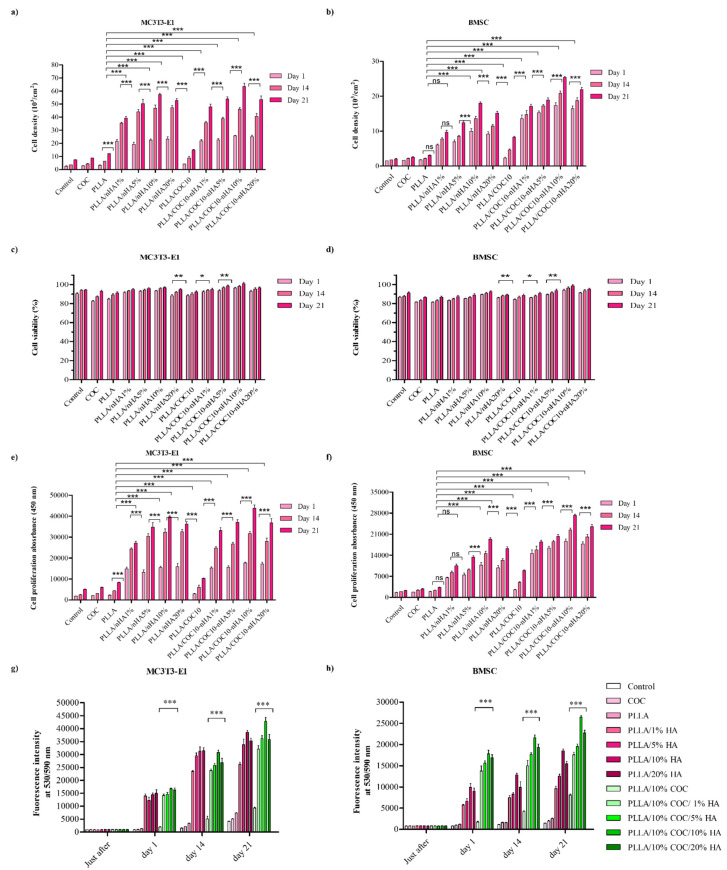
Cell culture data of PLLA and PLLA/COC10: (**a**) MC3T3-E1 density and (**b**) BMSC density. (**c**) MC3T3-E1 viability and (**d**) BMSC viability. (**e**) MC3T3-E1 proliferation rate (OD) and (**f**) BMSC proliferation rate (OD). (**g**) MC3T3-E1 fluorescence intensity and (**h**) BMSC fluorescence intensity. For days 7, 14, and 21, the results were statistically significant. Nanocomposites had nHA in 1 wt%, 5 wt%, 10 wt%, and 20 wt%. (Error bars: ±SD, * *p* < 0.05, ** *p* < 0.01, and *** *p* < 0.001).

**Figure 8 polymers-13-03865-f008:**
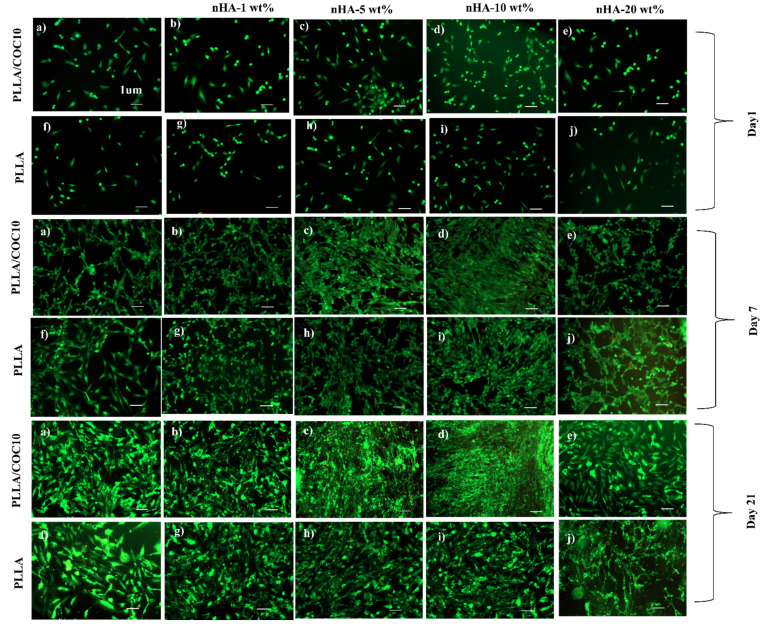
LIVE/DEAD assay for the vitality of MC3T3-E1 pre-osteoblasts cultivated on polymer, PLLA/COC10-nHA, and PLLA/nHA over time. The scale bar is 1 µm in all the images. Nanocomposites had nHA in 1 wt%, 5 wt%, 10 wt%, and 20 wt%. On the surface of the PLLA/COC10-nHA, cells adhered more as compared to PLLA/nHA at day 1 (**a**–**j**). The cells of day 14 (**a**–**j**) were building an interconnected network. Multilayer cells were found on day 21 (**a**–**j**) and demonstrated good cytocompatibility.

**Figure 9 polymers-13-03865-f009:**
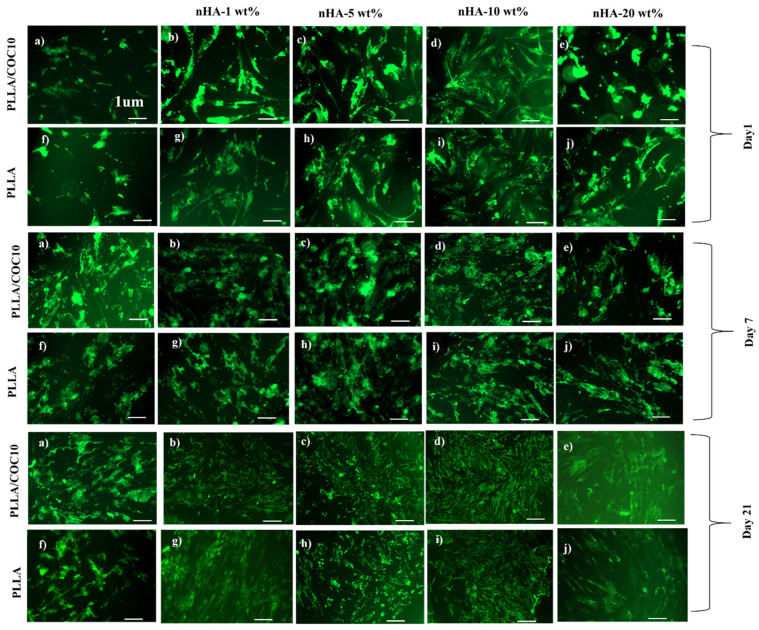
LIVE/DEAD assay BMSC grown on PLLA/COC10-nHA and PLLA/nHA at various times were tested. The scale bar is 1 µm in all the images. Nanocomposites had nHA in 1 wt%, 5 wt%, 10 wt%, and 20 wt%. PLLA/COC10-nHA on day 1 (**a**–**j**) represented cell adhesion better than PLLA/nHA. Day 14 (**a**–**j**) cells were proliferating, whereas multilayer cells were proliferating at an accelerated rate on day 21 (**a**–**j**).

**Figure 10 polymers-13-03865-f010:**
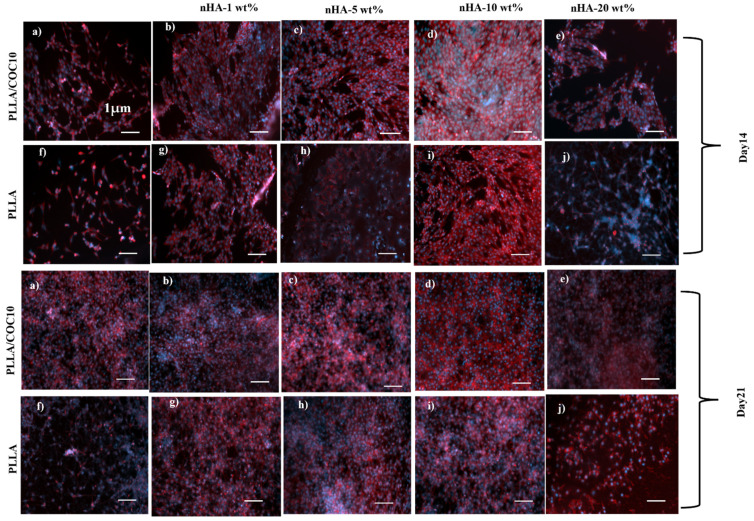
F-actin and DAPI staining in pre-osteoblast cells cultivated on PLLA/COC10-nHA and PLLA/nHA nanocomposites on week 2 (14 days (**a**–**j**)) and week 3 (21 days (**a**–**j**)). The scale bar is 1 µm in diameter in all the images. Nanocomposites had nHA in 1 wt%, 5 wt%, 10 wt%, and 20 wt%. The color of F-actin was red, while the fluorescence of DAPI-labeled nuclei was blue. The DAPI staining revealed the morphology of the nuclei. A large number of pre-osteoblast cell nuclei suggested that the PLLA/COC10-nHA were cytocompatible with the cell line. The presence of actin filaments in cells indicates that they were alive. Actin filaments revealed a complex cytoskeleton with well-defined direction and connectivity in cells.

**Figure 11 polymers-13-03865-f011:**
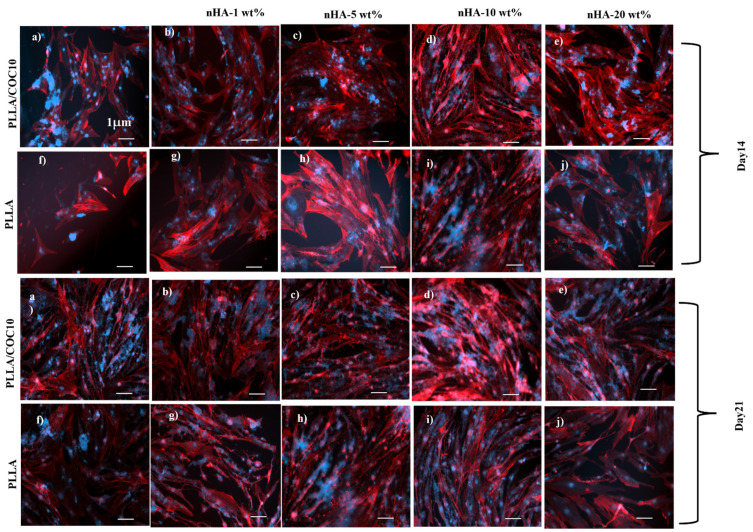
F-actin (cytoskeleton) and DAPI (nucleus) staining in BMSC cultivated on PLLA/COC10-nHA and PLLA/nHA nanocomposites on week 2 (14 days (**a**–**j**)) and week 3 (21 days (**a**–**j**)). The scale bar is 1 µm in diameter in all the images. Nanocomposites had nHA in 1 wt%, 5 wt%, 10 wt%, and 20 wt%. The DAPI staining revealed the morphology of the nuclei. A large number of BMSC nuclei suggested that the PLLA/COC10-nHA were cytocompatible with the cell line. Actin filaments revealed a complex cytoskeleton with well-defined direction and connectivity in cells.

**Figure 12 polymers-13-03865-f012:**
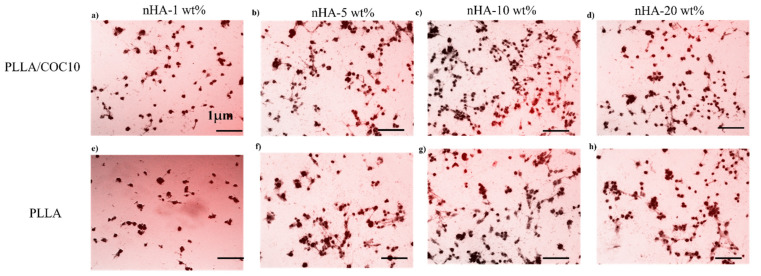
Alizarin red S staining was used to analyze the mineralization of MC3T3-E1 cells for PLLA/COC10-nHA (**a**–**d**) and for PLLA-nHA (**e**–**f**). MC3T3-E1 cells were grown for 14 days, then stained with alizarin red S to identify mineralized nodules after 14 days. Scale bar: 1 µm in all the images. Nanocomposites had nHA in 1 wt%, 5 wt%, 10 wt%, and 20 wt%. After three different experiments, the data is represented.

**Figure 13 polymers-13-03865-f013:**
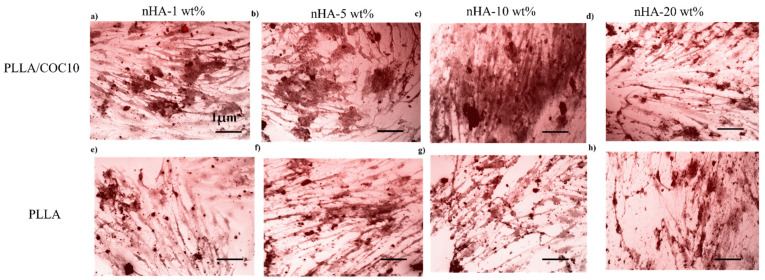
Alizarin red S staining was used to analyze the mineralization of BMSC for PLLA/COC10-nHA (**a**–**d**) and for PLLA-nHA (**e**–**f**). BMSC were grown for 14 days, then stained with alizarin red S to identify mineralized nodules after 14 days. Scale bar: 1 µm in all the images. Nanocomposites had nHA in 1 wt%, 5 wt%, 10 wt%, and 20 wt%. After three different experiments, the data is represented.

**Table 1 polymers-13-03865-t001:** Presenting the thermal parameters, such as *T_g_*, *T_m_*, Δ*H_m_*, and Δ*H_c_*, values of the PLLA/COC10-nHA and PLLA/nHA nanocomposites.

Nanocomposites	First Heating Scan		Cooling Scan	Second Heating Scan						
	*T_g_* ^1^	*T_m_* ^1^	*T_c_*	*T_g_* ^2^	*T_cc_*	*T_m_* _1_ ^2^	*T_m_* _2_ ^2^	Δ*H_m_*	Δ*H_c_*	Percentage Crystallinity (χ_%_)
PLLA/COC10	45.69	178.25	102	62	103.72 157.54	175.22	-------	50	18	37.94
PLLA/COC10/nHA 1 wt%	43	142	-------	43.7	103.89	129.85	140.68	18.9	−17.42	39.15
PLLA/COC10/nHA 5 wt%	45	142	-------	44.9	108.91	136.55	145.71	18.55	−18.26	41.35
PLLA/COC10/nHA 10 wt%	45	144	-------	45.73	108.92	137.38	145.70	18.48	−18.32	43.63
PLLA/COC10/nHA 20 wt%	46	144.7	-------	45.7	105.74	131.86	142.20	11.03	−11.15	29.58
PLLA	68	176	115	60	-------	178	-------	32	0	34.15
PLLA/nHA 1 wt%	48	144.3	-------	45.01	108.69	137.15	143.56	14.33	−14.56	31.143
PLLA/nHA 5 wt%	48	144	-------	46.6	108.30	134.32	142.48	13.46	−14.78	31.72
PLLA/nHA 10 wt%	48	143	-------	46.5	113.35	139.32	145.32	15.07	−13.23	33.55
PLLA/nHA 20 wt%	45	145	-------	44.5	97.77	127.57	140.40	10.07	−11.23	28.41

Here *T_g_*^1^ is the *T_g_* found in first heating while *T_g_*^2^ is the *T_g_* in second heat. *T_m_*^1^ is the melting temperature in first heat whereas *T_m_*_1_^2^ and *T_m_*_2_^2^ are first and second melting point peaks in second heat.

**Table 2 polymers-13-03865-t002:** Presenting the young modulus of PLLA/COC/10-nHA and PLLA-nHA nanocomposites.

Nanocomposite Composition	Young Modulus of PLLA/nHA	Young Modulus of PLLA/COC10-nHA
	(MPa)	(MPa)
Polymer	571.65 ± 2.8	843.0 ± 4.2
nHA1wt%	88 ± 5.4	107 ± 3.5
nHAwt5%	104 ± 4.5	263 ± 6.1
nHAwt10%	110 ± 6.2	278 ± 4.9
nHA20wt%	245 ± 3.5	289 ± 3.5

**Table 3 polymers-13-03865-t003:** Presenting the strain % of PLLA/COC/10-nHA and PLLA-nHA nanocomposites.

Nanocomposite Composition	Elongation at Break PLLA/nHA	Elongation at Break PLLA/COC10-nHA
	Strain %	Strain %
Polymer	7.84 ± 1.24	33.87 ± 3.2
nHA1wt%	12.197 ± 2.34	17.14 ± 2.6
nHA5wt%	10.37 ± 1.98	14.57 ± 1.4
nHA10wt%	5.56 ± 1.2	12.76 ± 1.7
nHA20wt%	2.62	6.49

**Table 4 polymers-13-03865-t004:** Presenting the maximum stress of PLLA/COC/10-nHA and PLLA/nHA nanocomposites.

Nanocomposite Composition	UTS of PLLA/nHA	UTS of PLLA/COC10-nHA
	Maximum Stress (MPa)	Maximum Stress (MPa)
Polymer	21.09 ± 2.45	24.4 ± 1.8
nHA1wt%	12.74 ± 1.34	10.29 ± 0.9
nHA5wt%	11.75 ± 1.03	11.5 ± 1.01
nHA10wt%	8.54 ± 1.2	9.41 ± 0.56
nHA20wt%	2.3 ± 1.3	7.25 ± 1.34

**Table 5 polymers-13-03865-t005:** Diameter of zone of inhibition of blends.

Nanocomposites	Diameter of Inhibition (mm)			
	*Escherichia coli* Gram (−)	*Pseudomonas aeruginosa * Gram (−)	*Staphylococcus aureus * Gram (+)	*Listeria monocytogenes * Gram (+)
PLLA/COC10-nHA 1 wt%	ns	ns	ns	ns
PLLA/COC10-nHA 5 wt%	ns	8.45 ± 0.3	ns	6.20 ± 0.10
PLLA/COC10-nHA 10 wt%	ns	10.39 ± 0.65	15.34 ± 0.11	14.73 ± 0.19
PLLA/COC10-nHA 20 wt%	ns	13.29 ± 0.43	8.21 ± 0.32	12.68 ± 0.43

**Table 6 polymers-13-03865-t006:** Description includes studies for bone-tissue engineering scaffolds to improve PLLA/HA nanocomposites.

Reference	Scaffold	Fabrication Procedure	Thermal or Mechanical Properties	Cell Type	Test and Results
[71]	PLLA/PCL/HA	Electrospinning	0.002–2.99 MPa (Young’s modulus)	MC3T3-E1 osteoblast	PLLA/PCL fibers with aggregates of nanophased HA. Introduction of nHA increased mechanical properties. Aligned fibers had good tensil mechanical properties. Scaffold supported adhesion and proliferation of preosteroblast cells. Antibacterial tests against *Staphylococcus aureus* showed lower number of colony forming units (CFUs), when PLLA/PCL fibers are aligned.
[22]	Loofah + PLLA + HA	Dip coated with PLLA solution	Stiffness 18−30 MPa PLLA Tm 160–175 °C	SW-1353 chondrocytes	Mechanical properties are result of strong interaction of HA/PLLA and loofah interaction. Metabolic activity suggested non-cytotoxicity of the scaffold. Overall results showed potential as cartilage tissue-engineering scaffold.
[60]	HAP/PLLA/PGA	3D printing	Tm values of PLLA decreased from 179 °C to 174 °C	MG-63 human osteoblast-like cells	Scaffold degradation rate was increased from 3.3% to 25.0% for 28 days. Good cell adhesion and proliferation was observed. Bone defects were bridged in 8 weeks.
[72]	PLLA/PCL/HA	Electrospinning	-	human dental pulp stem cells	HA-induced hydrophilic properties and led to improved biodegradation of fibrous membranes. Cells showed improved adhesion and proliferation capacity on the PLLA-PCL-HA nanofibers treated with MSH compared to other groups (*p* < 0.05)
[58]	Chitosan/nano-HA/nano-silver particles	Freeze drying	-	Rat osteoprogenitor cells and human osteosarcoma cell line	Scaffolds were characterized using SEM, FTIR, XRD, swelling, and biodegradation studies. Scaffold showed antibacterial results with *S. aureus* and *E. coli*. Scaffold showed cytocompatiblity with rat osteoprogenitor cells and human osteosarcoma cell line.
[73]	PLLA/Coll/HA	Electrospinning	-	MC3T3-E1 mouse osteoblasts	Composite morphology, diameter, and biodegradability was investigated. In vitro studies with MC3T3-E1 showed adhesion, proliferation, differentiation, and mineralization of cells on different nanofibrous scaffolds.

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
