# Peer review of "Comparative Study of Crystallization, Mechanical Properties, and In Vitro Cytotoxicity of Nanocomposites at Low Filler Loadings of Hydroxyapatite for Bone-Tissue Engineering Based on Poly(l-lactic acid)/Cyclo Olefin Copolymer"

_polymers, 2021, doi:10.3390/polym13223865_

Round 1

Reviewer 1 Report

In addition to rearranging and substantially rewriting, the text the following should be considered. 

Experiment Part. P3 Line 136. "nanocomposite was air dried for 24 hr ... were dried in vacuum oven" The process of the preparation of the composites is different with the scheme shown in Figure.1.

P3 Line101. What is the COC commercial grade? And what is the molecular weight of COC?

P7 Line 294. " Hydrogen boding between the polymer and nanoparticles ..." Does the polymer mean PLLA?

Please read some relevant references as follows.
Ref.21 Nazir, Farzana; Iqbal, Mudassir; et al. Abrication of robust poly L-​lactic acid/cyclic olefinic copolymer (PLLA​/COC) blends: study of physical properties, structure, and cytocompatibility for bone tissue engineering, Journal of Materials Research and Technology, 2021, 13, 1732-1751. 

1.  Lu, John C.; Kumar, Vipin; Schirmer, Henry G. Exploratory experiments on solid-​state foaming of PLA films and COC​/LDPE multi-​layered films,Annual Technical Conference - Society of Plastics Engineers, 2008, 66th, 1897-1901. 

2. Zakia Riaz, Ahmad Nawaz Khan, et al. Structure-properties relationships of novel cyclic olefinic copolymer/poly(L-lactic acid) polymer blends, Journal of Materials Research and Technology, 2020,9(4): 7172-7179.

Reviewer 2 Report

The manuscript is "

 Comparative Study of Crystallization, Mechanical properties, and In-vitro cytotoxicity of Nanocomposites at Low Filler Loadings for Bone Tissue Engineering based on Poly (L-lactic 4 acid)/Cyclo olefinic Copolymer ".

General comments:

  1. Please indicate the full name when the abbreviation appears for the first time
  2. Sample code “PC10-20HA” in Table1 is mistake
  3. The order of the Figure and Table, please follow the order of appearance in the article, such as Fig. 4 and 5
  4. Figure2, 12 and13 were prepared in low quality.
  5. Line 304, double ‘k’, line 390, Table 3 and line 612 Figure 9e and f are error.
  6. There is no h in Figure 9.
  7. The wettability of the material surface should be measured.
  8. How to measure the cell density between different materials in Line 667 to 688?
  9. There are no obvious differences for mineralization of MC3T3-E1 cells in Fig. 14, and what is the role of COC in this article? Can't see the benefits of COC on the mineralization of bone tissue engineering.
  10. The structure of the entire article is confusing and difficult to read. Please reorganize.
  11. Too many picture, figure and table support similar or same results, could be concise.

Reviewer 3 Report

In this work, nano hydroxyapatite (nHA) is added in lower filler loadings (1, 5, 10, 20 wt %) in PLLA/COC blend to obtain PLLA/COC-nHA scaffolds for bone tissue engineering. Further, structure activity relationship PLLA/COC10-nHA nanocomposites in compared with PLLA-nHA nanocomposites has been systematically studied. However, the following issues should be addressed.

1)    The nanocomposites PLLA/COC-10-nHA was fabricated by mixing HA and PLLA/COC directly. How to ensure the interface contact between organic PLLA/COC materials and inorganic HA materials. In the figure 2, the images of SEM showed that a large amount of HA agglomeration on the surface.

2)    In the figure 6a, the authors presented a stress curve of different nanocomposites. Please use corresponding icons to denote PLLA/nHA and PLLA/COC-nHA. For example, use solid triangle and hollow triangle to represent PLLA/nHA and PLLA/COC-nHA respectively. Please Avoid using too many colors in one single figure. It is difficult to distinguish.

3) In figure 6, the maximum strain with the increase of HA concentration, but the Young's modulus decreases with the increase of HA. Please explain the reasons

4) In figure 7b, the meaning of the ordinate is not clear, “degreation” may indicate remaining mass of nanocomposites.

5) Figure 8 is too blurry to see at all, and is not suitable for publication as a paper.

Author Response

Comments and Suggestions for Authors

Reviewer 3 Round 1

In this work, nano hydroxyapatite (nHA) is added in lower filler loadings (1, 5, 10, 20 wt %) in PLLA/COC blend to obtain PLLA/COC-nHA scaffolds for bone tissue engineering. Further, structure activity relationship PLLA/COC10-nHA nanocomposites in compared with PLLA-nHA nanocomposites has been systematically studied. However, the following issues should be addressed.

Comment: 1)    The nanocomposites PLLA/COC-10-nHA was fabricated by mixing HA and PLLA/COC directly. How to ensure the interface contact between organic PLLA/COC materials and inorganic HA materials. In the figure 2, the images of SEM showed that a large amount of HA agglomeration on the surface.

Response:  We performed the probe sonication for the preparation of the nanocomposites to get the maximum homogeneous mixing of the HA in PLLA/COC10. PLLA/COC10 clear solution in chloroform was probe sonicated for half an hour with nHA dispersed in chloroform. COC helped in minimizing the agglomeration which was unavoidable in PLLA-nHA.

Comment: 2)    In the figure 6a, the authors presented a stress curve of different nanocomposites. Please use corresponding icons to denote PLLA/nHA and PLLA/COC-nHA. For example, use solid triangle and hollow triangle to represent PLLA/nHA and PLLA/COC-nHA respectively. Please Avoid using too many colors in one single figure. It is difficult to distinguish.

 Response: In the figure 6a, corresponding icons to denote PLLA/nHA and PLLA/COC-nHA have been used and new figure has been added in the manuscript.

Comment: 3) In figure 6, the maximum strain with the increase of HA concentration, but the Young's modulus decreases with the increase of HA. Please explain the reasons

Response: young’s modulus in case of PLLA-HA decreased because of agglomeration while in case of PLLA/COC10-nHA young’s modulus values increased. Thus, making our hypothesis strong that in ternary system such as PLLA/COC10-nHA addition of COC has not only fine-tuned the structure and thermal properties but also enhanced the mechanical properties. We assume that COC chains are not only reinforcing the polymer matrix but are also involved in the better dispersion of nHA in polymer matrix (as compared to PLLA-nHA) as corroborated by SEM.

Comment: 4) In figure 7b, the meaning of the ordinate is not clear, “degradation” may indicate remaining mass of nanocomposites.

 Response: In figure 7b, “degradation” has been replaced by remaining mass % of nanocomposites.

Comment: 5) Figure 8 is too blurry to see at all, and is not suitable for publication as a paper.

Response: We have removed the Figure 8 and data is described in Table.

Round 2

Reviewer 1 Report

There were errors in English grammar, spelling and sentence structure need to be revised. 

Author Response

All the grammatical mistakes have been rectified in the revised version and highlighted in yellow.

Reviewer 2 Report

The author has fully explained and corrected

Author Response

thanks to the reviewer.